# Experiences of accessing primary care by those living with long Covid in New Zealand: A qualitative analysis

**Sarah Rhodes**⬤*, **Christina Douglas**

School of Physiotherapy, University of Otago, Dunedin, New Zealand

* sarah.rhodes@otago.ac.nz

## Abstract

### Background

Long Covid is the persistence of symptoms beyond 12 weeks following acute Covid-19 infection. It is estimated to affect one in ten people and can be extremely debilitating. With few publicly funded long Covid clinics, most people rely on primary care providers as a first point of contact. There is currently limited understanding of the experience of accessing primary health care by adults living with long Covid in New Zealand.

### Purpose

To explore the experiences of accessing primary health care by adults living with long Covid.

### Methods

A narrative inquiry approach was used to capture participants lived experiences of accessing primary health care. Zoom interviews and discussions were conducted with study participants. The automatically generated transcripts were reviewed and corrected, and the collated data were analysed using Braun and Clarke's thematic analysis.

### Results

Eighteen people participated in the interviews. Codes were identified and, through an iterative process, themes were generated, reviewed, and named. The seven themes included lack of upskilling of primary care staff; let down by the Government; self-advocacy and its cost; and throwing money at it.

**Data availability statement:** All relevant data are within the manuscript and its Supporting Information files.

**Funding:** This research has been funded by an Activation Grant from the Health Research Council of New Zealand received by SR. The funder had no role in the study design, data collection and analysis, decision to publish or preparation of the manuscript. https://www.hrc.govt.nz/grants-funding/role-ethics.

**Competing interests:** The authors have declared that no competing interests exist.

## Conclusion(s)

The picture painted by participants was bleak with a sense that the world had moved on from Covid-19 and left them behind, with some experiencing a lack of support in primary health care. Reducing the likely long-term health and economic burden of long Covid requires targeted investment and action by Government at every level, along with better utilisation of the allied health workforce in primary care.

## Introduction

Long Covid, or post-Covid condition, is the persistence of symptoms three months following an initial SARS-CoV-2 infection and can be severely debilitating [1]. It is estimated to affect up to 45% of those who develop a Covid-19 infection [1] and the risk of long Covid persists with each subsequent Covid infection [2]. Although Covid-19 is no longer considered a public health emergency by the World Health Organization [3], long Covid presents an ongoing and complex challenge to affected individuals, their families and health systems [4].

Although there is currently no cure for long Covid, there are ways of managing the condition. Studies show that various rehabilitation strategies [5–7] can result in a reduction in long Covid symptoms; as well as demonstrating some benefit from behavioural interventions [8]. Globally, both primary care providers and hospitals have developed services to provide long Covid management, with a range of options available, including in the United States (US) [9], the United Kingdom (UK) [10] and Australia [11]. However, research suggests that general practitioners (GPs) in some countries are impacted by the challenges of this new condition with its complex and varied clinical presentation [12], coupled with time-limited appointments [13].

In New Zealand (NZ), the responsibility for managing long Covid sits in primary care, as highlighted by the Ministry of Health guidelines [14]. However, with a lack of funding and available resources to date, there is currently only one publicly available clinic, with a scattering of private providers across the country. With the primary care system under increasing pressure, and some general practices closed to enrolments due to underfunding, staff burnout and an ageing workforce, dedicated support for those living with long Covid looks unlikely [15]. Most people living with long Covid in NZ rely on their GP as a first point of contact [12]. However, there is a prevailing sense that NZ has moved on from the pandemic and that some of those living with long Covid have been overlooked, as is the case elsewhere [16]. This is particularly concerning for Māori and Pacific peoples, who already experience inequitable access to health care [17]. This concern about being left behind is echoed in the US [18] with concerns that minority communities are not well represented in long Covid research. Currently, little is known about patients' experiences of accessing health care for long Covid or the quality of care received in NZ. The aim of our study was to explore the experiences of people living with long Covid in accessing primary care within NZ.

## Methods

This project was approved by the University of Otago Human Ethics Committee (H23/003). All procedures performed in studies were in accordance with the 1964 Helsinki declaration and its later amendments or comparable ethical standards. The study is reported in accordance with the Consolidated criteria for reporting qualitative research (COREQ) checklist [19] (S1 Appendix).

## Participants

Eligible participants were anyone aged 18 or over with symptoms that met the World Health Organization definition of long COVID [20]. The only exclusion criteria was anyone who couldn't conduct the interviews or discussions in English. The aim was to include 15–20 participants as this was deemed to be large enough to provide sufficient depth to identify recurrent patterns in the data. Given that the interviews lasted at least an hour, were conversational and adaptive in nature, and time was spent building rapport with participants, this provided confidence of the sufficiency of the data, and a high degree of information power [21].

Participants were recruited through convenience sampling via the NZ long haulers Facebook group, which has since become Long Covid Support Aotearoa (LCSA). A study invitation (S2 Appendix) was posted on the group page, and potential participants were asked to contact the PI if they were interested in participating. If interested and eligible, participants were emailed a study information sheet and consent form. Recruitment occurred between 1st June 2023 and 1st February 2024. This online community support group comprises over 2,600 members who are all people living with Long Covid; they are well informed and have been instrumental in raising awareness of long Covid within NZ. Written informed consent was obtained from each participant.

Drawing on the story telling tradition, the research used a narrative inquiry methodology, which sought to adopt a person-centred approach to give voice to each individual participant's long Covid journey [22], as a means of understanding their lived experiences in depth. This aligns with the ontological perspective of relativism, that posits that there is no one single reality. People acquire knowledge and understanding through their different contexts and lived experiences. A strength of the narrative inquiry approach is that it provides flexibility in exploring layers of meaning both within individual stories and across multiple stories, allowing recognition of the uniqueness of each person's reality, as well as the shared elements. This creates a richness to the data and explores the multiple dimensions of the journey, including the emotional aspects, associated with the impact of chronic illness. The aim was to provide a supportive and empowering space for these stories to be heard [22]. One potential limitation is that some participants can find the telling of their story both poignant and tiring [23]. This was anticipated and participants were aware that they could stop their story and/or take breaks from the interview/discussion group as they saw fit and without explanation.

The researchers each completed a reflexivity statement to identify their own reality and reflect on the individual values and beliefs they may hold. Both researchers have a professional background as physiotherapists. Neither has experienced long Covid personally. Both have worked with various patients managing long term conditions, providing a better understanding of how these can impact a person's life. CD has no prior experience of working with patients with long Covid or doing research in this area. SR has done long Covid advocacy work and, as a result, holds an underlying assumption that those living with long Covid in NZ are not well supported. Being reflexive allowed for better recognition of assumptions such as these, and to acknowledge how their subjectivity might shape the interview process and the subsequent analysis of transcripts [24], through their own knowledge and perspectives of long Covid. This was to help reduce any personal biases that might impact the interpretation of the data and enhance credibility and dependability of the findings.

Participants were invited to take part in a group or individual Zoom session based on their preference. Participants were given the option to provide supplementary written information where the online sessions were considered too fatiguing. The aim was to recruit at least twelve people with lived experience. Each group session lasted approximately one hour.

The individual interviews and discussions explored participants' experiences of living with long Covid. Part of this journey included experiences of accessing primary care for their condition. The focus was on providing space for peoples' stories to be heard and deriving meaning from the collective experience. Zoom interviews began with an introduction to the study [S3 Appendix] and a few loosely structured questions were used to encourage discussion about participants' experiences. [S4 Appendix] The interview questions were developed by the research team and piloted by two lay people. Feedback was sought regarding ease of reading and flow of questions. Two questions were reworded as a result. The zoom sessions were facilitated by a female member of the research team (SR) who holds a PhD and has previous experience of conducting qualitative interviews.

The aim was to recruit a minimum of 15 participants, which was considered a sufficient sample size to obtain data rich and varied enough to answer the research question and advance knowledge in this area, based on other studies [25]. The concept of information power was used rather than data saturation, which is a questionable concept in reflexive thematic analysis [25]. Information power was considered in the use of semi-structured interviews to collect rich, meaningful data and by allowing the researcher (SR) to explore concepts in more detail through probing questions to facilitate quality dialogue. Additionally, consideration was given to sample specificity and the means of data analysis used to further enhance the information power of the study [26].

## Procedures

Participants' demographic data were collected via email. Data were anonymized and collated before being stored on the primary researcher's password protected computer.

## Data collection

Zoom interviews and discussions were recorded using the Zoom recording function, which generated a transcript that was temporarily stored in the Cloud. Once a transcript recording was completed and saved to the Cloud, the primary researcher received an automatically generated email. The transcript was then downloaded and saved on a password protected computer. The transcript was reviewed against the recording for accuracy, and the original recording was deleted. Trustworthiness was addressed through member checking [27]. Transcripts were emailed to participants to ensure they were an accurate reflection of their views and for them to make any amendments to ensure credibility of the data, after which the transcripts were deidentified.

Where participants consented but didn't have the energy to participate in a Zoom session, they were given the opportunity to provide their answers to the questions in their own time and email them to the research team.

## Data analysis

The transcripts and email responses were analysed using Braun and Clarke's six step reflexive thematic analysis [28]. This complimented the narrative inquiry approach to data collection, since it is an inductive approach which serves to identify and interpret patterns in the data, enabling development of a collective understanding of lived experiences. Data analysis was initially undertaken using manual methods through annotating printed deidentified transcripts. No coding software was used. Initially, both researchers (SR and CD) undertook familiarisation with the transcripts through repeated reading, making notes and developing preliminary codes. This was followed by a more robust coding process, identifying and sorting data into meaningful groups. Codes were then added into an Excel spreadsheet, where they could be easily separated or combined as coding is further developed and refined. Once codes were identified, reviewed and agreed upon, they were analysed and grouped to generate larger over-arching themes. [S5 Appendix] Definitions for each theme can be found in S6 Appendix. There was some representation of codes within more than one theme, which reflects the complexity of dialogue, when participants address the interrelationship between several complex ideas simultaneously. This is not explicitly encouraged or discouraged in reflexive thematic analysis.

Themes were further refined through discussion and named to reflect their key quality. Write up involved reporting on the story the data reflected to present the findings in a meaningful way.

## Results and discussion

Eighteen participants consented to be in the study. Sixteen were interviewed; the interviews entailed two groups of five people, one group of three people, and three individual interviews. Two participants provided written responses as a zoom interview was considered too tiring. Participant characteristics can be seen in Table 1. Geographical location was categorized using Geographical Classification for Health categories: U1, U2, R1, R2 and R3, with U1 being the most urban and R3 being the most rural [29]. U1 and U2 are based on population size, and R1 – R3 are based on drive time to their closest major, large, medium, and small urban areas.

One participant responded with suggested amendments to their transcript.

Seven themes were developed: gaslighting and validation; lack of support and unmet need; inequity of available care; lack of upskilling of primary care staff; let down by the Government; self-advocacy and its cost; and throwing money at it.

### Theme 1: Gaslighting and validation

Gaslighting is defined as manipulating someone into questioning their version of reality [30]. Our findings suggest that, for this cohort of people with lived experience of long Covid, there were several barriers to accessing care, including health professionals' attitudes to the condition (Theme 1 in S5 Appendix). Participants highlighted the fact that often their experiences of accessing primary care resulted in prejudice; primarily not being believed and being made to feel like they were wasting health professionals' time or being labelled as someone who didn't contribute to society.

*One of the problems for people who've got long Covid that always makes me want to cry is that people don't believe us.* [Participant 3]

*Someone who, effectively, my doctor's term would be 'a weight on society.'* [Participant 13]

This lack of validation of symptoms, and feeling dismissed by their doctors, has been an issue for many people presenting with long Covid in primary care, as reported by others living with the condition globally, including the UK and US [14,30–33].

This is despite greater understanding of the condition over time and a large body of literature confirming the physiological impacts [34]. This medical gaslighting may be partly due to the reliance on medical testing which does not always capture the physiological changes associated with long Covid, as well as an absence of established biomarkers to confirm the diagnosis [34]. Gaslighting of patients with 'invisible' symptoms is not new: it has been commonly experienced by the those living with conditions such as chronic pain for many years [35].

Although most participants reported feeling stigmatized, some acknowledged that their experiences with health professionals were reassuring and helpful, reflecting the variation in attitudes within NZ. There was a sense that listening, acknowledging and being empathetic were valued, even in the absence of clear treatments.

*That validation is just so important like, I went to a new medical practitioner,..., last week. And the outstanding thing to me was, he said, 'Oh, okay, how does long Covid affect you?' And he talked about it as if it was perfectly normal to not be able to remember things.* [Participant 13]

Where health practitioners did listen and validate, this had a positive effect on the patient experience. Several studies have proposed that health professionals' attitudes to long Covid present the greatest barrier to care [36–38]. Where there is continuity of care and a trusted relationship develops between patient and clinician, this can be game changing for the patient [14,38].

### Theme 2: Lack of support and unmet need

Participants raised the issue of facing barriers in the form of health professionals' lack of supportive action. Several participants outlined the challenge of getting support in the form of long waiting times, no onward referral, and a lack of equipment or practical solutions, resulting in needs not being met (Theme 2 in S5 Appendix).

**Table 1. Participant characteristics.**

| Characteristics | Number (percentage) |
|---|---|
| **Gender** | |
| Male | 3 (17) |
| Female | 13 (72) |
| Gender diverse | 2 (11) |
| **Ethnicity*** | |
| European | 15 (83) |
| Māori | 4 (22) |
| Pacific Peoples | 3 (17) |
| Asian | 0 (0) |
| Middle Eastern/Latin American/African | 1 (6) |
| Other Ethnicity | 0 (0) |

*Some participants identified as more than one ethnicity*

| Characteristics | Number (percentage) |
|---|---|
| **Age** | |
| 18-29 | 3 (17) |
| 30-39 | 2 (11) |
| 40-49 | 4 (22) |
| 50-59 | 4 (22) |
| 60-69 | 4 (22) |
| 70+ | 1 (6) |
| **Urban/rural** | |
| Urban U1 | 7 (39) |
| Urban U2 | 5 (28) |
| Rural R1 | 3 (17) |
| Rural R2 | 2 (11) |
| Rural R3 | 1 (6) |
| **Region** | |
| Auckland | 3 (17) |
| Waikato | 1 (6) |
| Gisborne | 1 (6) |
| Manawatū-Whanganui | 3 (17) |
| Wellington | 5 (28) |
| Nelson | 1 (5) |
| Marlborough | 1 (6) |
| Canterbury | 1 (6) |
| Otago | 2 (11) |
| **Education** | |
| Doctoral degree | 4 (22) |
| Master's degree | 7 (39) |
| Postgraduate diplomas and certificates, bachelor honours degree | 1 (6) |
| Bachelor's degree/graduate diplomas and certificates | 3 (17) |
| Diplomas | 2 (11) |
| Certificates | 1 (6) |

*(Continued)*

**Table 1.** (Continued)

| Characteristics | Number (percentage) |
|---|---|
| **Employment** | |
| Full time | 2 |
| Part time | 5 |
| Unemployed | 0 |
| Retired | 3 |
| Student | 1 |
| Unable to work currently | 7 |

*I can't get any assistance. So, the doors keep closing everywhere.* [Participant 18]

*Go to the doctor and they say 'Oh no, the waiting list is a year* [to see a specialist] [Participant 13]

Additionally, the lack of joined up care made wider support difficult to come by, with several participants underlining the constraints of the existing system.

*I struggled a lot with work and income around long Covid because they require all disabilities to have an end date and no one can give an end date when I'm going to stop having long Covid.* [Participant 4]

*Even people who are eligible for home help can't get home help because of the way they fund the carers.* [Participant 1]

This experience of unmet need is not limited to NZ, with other studies highlighting delays in referral or treatment, as well as a lack of practical options offered [39,40]. An online survey of 10,462 adults with long Covid in the Netherlands, reported 34% found the health professional they accessed was unable to help them and 25% had a long waiting time to access help [41]. Two systematic reviews of those living with long Covid have highlighted available services as being slow, underdeveloped and not helpful [42,43]. In the US, patients in primary care reported barriers at every level of the healthcare system [44]. Although there is a dearth of evidence regarding how GPs are coping with the management of long Covid in NZ, studies in Germany [45] and Belgium and Malta [46] reflect the challenges GPs find in diagnosing and managing patients with long Covid, in terms of time, lack of biomarkers, lack of scientific knowledge and the complexity of the condition. Overall, our study participants perceived lack of support left them feeling constrained by existing health system structures and that the system wasn't fit for purpose. This is mirrored in other research where US participants reported feeling that the system was set up for people to fail at obtaining help [39].

## Theme 3: Inequity of available care

Health professionals' lack of knowledge, lack of action and limited resources, contributed to a variation in care received across different locations and primary care providers, including evidence of urban-rural variation (Theme 3 in S5 Appendix).

*It's like so piecemeal, is the way I would describe it at the moment. You're really lucky when you find someone good.* [Participant 8]

*I'm just putting my rural context on that because there's nothing available here.* [Participant 1]

Sometimes the inequities were present even in the same city, which appeared to reflect a variation in primary care providers' knowledge of both the condition and of the available referral pathways for long Covid related symptoms.

*[My GP's] they haven't known what … to do with me. I know someone who's had long Covid since she had Covid at the start of March this year; so only months. And she's already got a POTS diagnosis. She's already seen cardiology. She's in the same city as me. She's as sick as I am but she can get all of that stuff. I still can't even get referred for a tilt table test.* [Participant 4]

One participant noted the variation in the care she had received compared with those who had more visible conditions, such a traumatic injuries.

*I almost wish I'd been in a car accident because then at least I would have some support.* [Participant 2]

Participants also commented on the fact that those with other visible health conditions, such as cardiovascular disease or type 2 diabetes, had better access to health services with established care pathways [47]. The invisible nature of long Covid appears to have contributed to the lack of available care [34].

However, even differences between different invisible conditions were highlighted with some clinical presentations appearing to be more accepted and supported than others.

*If you don't have a brain injury per se, but your brain is being affected by a virus, there just seems to be this silence around it.* [Participant 9]

The paucity of care received by those living with invisible illnesses, such as myalgic encephalomyelitis/chronic fatigue syndrome, has been well documented in the literature [48,49] and, to date, there are limited therapeutic options available [50]. This is further hampered by insufficient support for research funding in NZ, with patient donations proving the impetus for research funding [51]. However, some invisible conditions, such as concussion, do have well established treatment plans. Despite increasing recognition of the overlap between the clinical presentation of long Covid and persistent concussion and calls in the literature to use post-concussion frameworks for those with long Covid [52], this appears to have gained limited traction in practice. Utilising existing skills to treat overlapping symptoms, as well as existing pathways as a starting point, may be one means of reducing inequity of care.

## Theme 4: Lack of upskilling of health care staff

The variation in services provided was perceived to be partly due to the lack of funding and resources to upskill health care staff in the assessment and management of long Covid (Theme 4 S5 Appendix). With fewer doctors entering general practice, the current primary care system in NZ is overwhelmed and unable to meet demand; existing GPs report feeling stressed and overworked, [53] leaving little time for upskilling. Participants wanted more explanation and guidance from the health professionals they saw regarding their condition.

*What I would love to be able to do is go to a medical professional and they could guide me through finding the right language to, to say what it is I am experiencing…* [Participant 14]

Other health professionals working in primary care were sometimes perceived as not having the skills to manage patients presenting with long Covid.

*She* [the HIP] *doesn't know what to do. I feel so sorry for her. She's I think, she's overwhelmed.* [Participant 18]

This led to concerns that some health professionals were not accessing the current recommendations for management of long Covid, despite the publication of Long Covid rehabilitation guidelines by the Ministry of Health [15].

*And there are still physios doing graded exercise therapy for people with long Covid. And, if you're going to have physio referrals, they need, physios need updating and educating as well.* [Participant 15]

This sense that their health providers were still learning about the condition and were experiencing uncertainty regarding how to manage patients echoes the experiences of patients attending a post-Covid recovery clinic in Ohio [54]. It is further supported by literature in NZ, which highlights a lack of confidence amongst primary care-based physiotherapists in managing long Covid [55]. The urgent need for education on long Covid for healthcare workers in NZ has been highlighted in the media by other researchers [56].

Experiences of staff having limited access to education and resources are also reflected in other study cohorts, such as in the UK [57]. Although there is some evidence of resources being developed for healthcare workers to support patients with long Covid in Canada [58], there does not appear to be a cohesive approach. Ultimately, dedicated workforce training on long Covid management, alongside clear signposting of credible online resources, is required to better support health professionals in primary care [59]. This issue has been raised by GPs in Germany, who highlighted the need for a uniform database with up-to-date information on diagnosis and treatment of long Covid [45]. In NZ, although there are advocates for the development of accessible specialist long Covid services and treatment pathways [60], there is no evidence that these are forthcoming.

**Theme 5: Let down by the Government**

This inability to access the necessary care, and the perceived lack of funding and support for staff in primary care to upskill, led to a feeling of being let down by the Government (Theme 5 S5 Appendix).

*Everyone goes 'oh the poor [GP] practice, you know, the poor practice, they are so under pressure, it's like, well stuff it actually! You know, this is where you're meant to be coping with us. The Government has said you are responsible for this chronic illness but they haven't got the resources to do it.* [Participant 8]

Additionally, there was disappointment and anger at what was regarded as a lack Government action.

*So it's greatly ironic that I'm actually here today and extremely angry because everything I planned for [in terms of policy] has been ignored by the public policy system.* [Participant 10]

*I mean, I feel totally [enraged] with the Government,…, But for the Government to say, to be so silent on long Covid. I mean, it's not just, it's illegal.* [Participant 11]

Long Covid is associated with increased health care use and substantial primary care costs [61]. There is a perceived lack of government action in NZ, in terms policy initiatives and additional funding to primary care, to support those living with long Covid. While Governments in other countries, such as the US and Australia, have established dedicated long Covid funding, the NZ Government appears to have washed its hands of the issue and is contributing nothing [51]. This is despite previous calls from NZ health professionals to provide funding to support those living with the condition [62]. With the NZ health system under increasing pressure, and an admission that the system is in crisis with high levels of unmet need [63], specific funding for long Covid is unlikely with Health New Zealand highlighting that there is no additional funding for investigation and treatment of long Covid [64]. The current situation is one of geographical variation and fragmented care, primarily offered by private providers. This mirrors experiences elsewhere. In Ontario, Canada, the lack of a cohesive long Covid strategy by government has led to a disjointed approach to patient care, which is now under threat [65]. However, this is not the case everywhere. In England, the NHS has invested £34 million on over 80 adult long Covid clinics [11]. Likewise, most top hospitals in the US provide some form of long Covid service, although there is variation in terms of what is offered [10].

In other places, the absence of clear government action has resulted in strong advocacy from the primary care sector, with organisations such as the Royal Australasian College of Physicians raising concerns about the closure of Australian long Covid clinics and appealing for government funding [66]. Studies have highlighted an urgent need for innovative, cost effective models of care to successfully meet the needs of patients living with long Covid [12]. Although potential models of care have been reviewed and acknowledged in a 2025 NZ Government briefing document [67] and recommendations made, there is no detail on how this might be achieved and still no change in practice.

**Theme 6: Self advocacy and its cost**

In the absence of readily available support, self-advocacy was a key feature of most participants' approach to accessing care. Some viewed this as a necessary part of the process to move things forward in getting the support they required. Perhaps, in part, due to the majority of the cohort having a higher educational background, they were very proactive and not afraid to ask for what they needed.

*My doctor wouldn't refer me for ages. So I basically got the clinic to contact her. I'd given them permission to look at my medical records. So I went over her head to do it.* [Participant 12]

However, others expressed their frustration and annoyance that it had to be this way.

*You know, like all that work, all that advocacy that I have to do, all of that in order to get them to listen is infuriating.* [Participant 9]

For others, self-advocacy represented them finding their own solutions to managing their condition or sending information to health professionals.

*I think we've all had to do a lot of investigation into stuff because when there is no one else that's had it and, you know, with my GP I was constantly sending her links; please read this, you know.* [Participant 1]

This need to find their own solutions came at a cost.

*I've got to read and learn myself about how I could then apply it to my schedule. But there is a huge energy cost to that.* [Participant 12]

The ongoing lack of support for those living with long Covid in NZ has been frequently highlighted by scientists in the NZ media [56,68,69]. However, patients seeking their own creative solutions to managing their condition is not unique to NZ [37]. Elsewhere, including the UK, others have been proactive in their engagement with the health system [43] through decisive action such as switching GPs, demanding referral to a long Covid clinic or asking for referral to specialist [14]. A further UK study reported that patients viewed advocating for themselves as an attempt to regain control in a situation of uncertainty [33].

This proactive approach to health care by patients has the potential to shape patient-clinician partnerships in the future. The notion of shared decision making is at the heart of person-centred care and acknowledges the patient's expertise about their condition [70]. However, nurturing therapeutic relationships is challenging in a system where GPs are over-stretched due to staff shortages, an ageing workforce and an increasing number of patients presenting with complex con-ditions [71]. One solution might be to better utilise the allied health workforce in the primary care sector [72], which aligns with the existing NZ Government priorities [73].

### Theme 7: Throwing money at it

The lack of recognised treatments for long Covid led to people seeking out alternative options; some without any support-ing evidence and often at a high financial cost.

*I've tried over 20 different forms of treatment and therapy.* [Participant 7]

*So she told me the name of the expert, …, so I made a private appointment to see him at $405. Just about killed me. But anyway, and he had nothing to offer. He had nothing to offer.* [Participant 3]

Participants acknowledged that they were prepared to try any form of alternative treatment in an attempt to alleviate symptoms.

*I've tried the alternative, you know. Hypobaric chambers, osteopaths, various different supplements, acupuncture, just in desperation to do something.* [Participant 2]

*You can spend an awful lot of money on alternative treatments…There's lots of things you could spend your money on and you'd still be going, I'm not sure if that's helping.* [Participant 8]

There was frustration that people with long Covid were ripe for exploitation due to the failure of health services to pro-vide for them.

*Online it is clear that desperate souls are open to being taken advantage of by alternative practitioners – our care should be clear in mainstream medicine so we are not open to this!* [Participant 18]

Although there is little available research to support this happening in other long Covid cohorts, one study highlighted the impact of people with long Covid engaging with dubious health sources online [14]. The issue of "quackery" in health care is not new, with a recent scoping review seeking to determine the reason for the development of quack medicine [74]. The authors' concluded that quack medicine is caused by a range of factors, including political, economic, sociocultural and psy-chological [75]. In the context of long Covid, it is apparent that this population are vulnerable due to the ongoing gaslighting some have experienced at the hands of medical professionals [14] and the lack of available treatments [34]. The resulting disappointment and despair [75] provides a perfect opportunity for exploitation by unscrupulous practitioners.

Insufficient information and health literacy both contribute to susceptibility to quackery [74]. Raising awareness of long Covid through a public awareness campaign, with a particular focus on those groups who are often marginalised by health care, might serve as a good starting point. This could include communicating where to access credible online resources, which could support some people in self-managing their condition. The need for better access to information has been highlighted by those living with long Covid in other places, such as the UK [75].

## Strengths and limitations

One of the key strengths of this study was the qualitative approach to data collection, which thoroughly captured patient experiences, resulting in richly detailed data, supporting the purpose of the study. This approach supported participants to share their stories, allowing for more nuanced perspectives than alternative data collection methods such as surveys. Member checking of transcripts ensured the data accurately reflected the patient voice. The varying methods of data collection (group, individual, written) supported wider inclusion and enabled access to a broader range of experiences by ensuring even those with highly disabling symptoms were able to participate. The use of group discussion for data collection was both a strength and a weakness. The strength lay in providing a forum for those with a shared experience which encouraged openness and supported deeper level discussion.

A potential limitation of data collection was the dominant voice of some participants within group discussions although this was largely mitigated by the facilitation skills of the researcher.

Another limitation of our study was that participants were all recruited from an online support group, which may introduce selection bias. Individuals who use such forums may have higher levels of health literacy. Additionally, those who seek support from these groups may do so because they have a higher symptom burden. Therefore, their views may not reflect the wider cohort of people living with long Covid. Another limitation was the lack of diversity within our sample in terms of educational background; most participants were highly educated and had a post graduate qualification. This suggests higher health literacy and greater likelihood of engaging with primary care services. Our cohort is likely to have been adept at advocating for themselves and able to successfully navigate the health system. This will have been reflected in the themes that were derived from our study, particularly the emphasis on self-advocacy. It is necessary to note that this is unlikely to be reflective of the wider cohort of people living with long Covid, such as Māori and Pacific Peoples who are underserved by the education system, therefore limiting transferability of the findings. Additionally, although our cohort included Māori and Pacific representation, none of the participants identified as Asian, and only 17% were male. All the afore-mentioned factors may have impacted our findings; they may not be representative of the wider population affected by long Covid and, therefore, cannot be generalised. However, it should be noted that females have a higher risk of developing long Covid than males [76].

Despite the inclusion of Māori and Pacific participants in the study, these participants' voices were not analysed independently of the wider group findings. This signifies a potential missed opportunity to better understand these perspectives in the context of the recognised health inequities within NZ.

## Conclusion

This is the first study to highlight the experiences of accessing primary care by a cohort of people living with long Covid in NZ. The picture painted by these participants is bleak with a sense that the world had moved on from Covid-19 and left them behind. Patients accessing the current system experience challenges, including gaslighting, unmet need, inequity of care and uncertainty amongst health providers regarding their condition. In response they are having to self-advocate which often comes at both a high personal and financial cost. Overall, there is a sense of being let down by the health system in NZ. This sentiment is reflected elsewhere, including in places where long Covid services were initially funded and are now being shut down, such as Australia [77] and the UK [78].

Despite the existence of long Covid for several years, and the NZ Ministry of Health providing clinical recommendations [15], these have not been updated since 2022. There is no appetite from Government to support the provision of public long Covid services with Health New Zealand clearly stating that there is no additional funding for long Covid [64]. This is despite the NZ Royal Commission Summary report on lessons learned from Covid-19 acknowledging that many people in NZ remain unwell due to long Covid [67]. This is in contrast to Australia, where the Government has allocated funds to both long Covid healthcare and research, as part of its National Post-Acute Sequalae of COVID-19 plan [79].

Perhaps unfairly, the burden of care in NZ has been placed on an already under pressure primary care system, and in the absence of any accompanying training or additional funding. It is expected that any additional clinical provision will be absorbed into existing caseloads. With the current pressure on the NZ health system, and the failure of Government to provide dedicated funding for long Covid several undesirable outcomes look increasingly likely: inadequate support for patients presenting with this complex and disabling condition, even greater pressure on the primary care system, and wider health disparities, particularly for Māori and Pacific peoples [67].

For NZ to effectively support those living with long Covid, there need to be a range of initiatives to ensure access to fit-for-purpose long Covid care in the public health system which is accessible to all. Government investment is needed at every level; in digital infrastructure, in training to upskill health staff and in expanding the allied health workforce. A lack of action risks increasing the health and economic burden of long Covid, as well as perpetuating health inequities. Long Covid is not going away and those representing patients, health professionals and researchers will continue to advocate for better support. It is time for the Government to listen, step up and take responsibility.

## Recommendations

For Government:

- Greater investment in primary care to support the management of long Covid

- Investment in digital infrastructure to reduce administrative barriers between primary and specialist care to create seamless coordinated care pathways

- Leverage the allied health workforce through embedding physiotherapists and other health professionals in established primary care services

- Create physiotherapy led long Covid clinics for those patients presenting with common symptoms such as post exertional malaise, postural orthostatic tachycardia syndrome and breathing pattern disorders

- Create a national online platform offering evidence-based standardised long Covid training for primary care practitioners

- Create a public awareness campaign for better delivery of credible information to support self-management of those living with long Covid where possible

  For primary care providers:

- Develop agile practice through increased use of remote models of care, in addition to face-to-face services, to reduce inequities

- Upskill through accessing available long Covid training

- Use health navigators to increase access to those living with long Covid

  For policy developers:

- Co-design and co-creation of future services with input from those living with long Covid

- Policies to support better collaboration between different government departments, such as Ministry of Health and Ministry of Social Development, to remove barriers to accessing services and ensure wrap around service provision

## Supporting information

**S1 Appendix. COREQ checklist.**
(DOCX)

**S2 Appendix. Study invitation.**
(DOCX)

**S3 Appendix. Zoom session introductory information.**
(DOCX)

**S4 Appendix. Zoom discussion questions.**
(DOCX)

**S5 Appendix. Codes and themes.**
(DOCX)

**S6 Appendix. Definitions of themes.**
(DOCX)

## Acknowledgments

We wish to acknowledge the School of Physiotherapy, University of Otago for providing access to resources and support. We also wish to acknowledge the participants who entrusted their stories to us and shared their journeys.

## Author contributions

**Conceptualization:** Sarah Rhodes.

**Data curation:** Sarah Rhodes.

**Formal analysis:** Sarah Rhodes, Christina Douglas.

**Funding acquisition:** Sarah Rhodes.

**Methodology:** Sarah Rhodes.

**Project administration:** Christina Douglas.

**Writing – original draft:** Sarah Rhodes.

**Writing – review & editing:** Sarah Rhodes, Christina Douglas.

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
