## [Decision Letter · Decision Letter 0]

15 Aug 2025

Dear Dr. Rhodes,

Thank you for submitting your manuscript to PLOS ONE. After careful consideration, we feel that it has merit but does not fully meet PLOS ONE’s publication criteria as it currently stands. Therefore, we invite you to submit a revised version of the manuscript that addresses the points raised during the review process.

We look forward to receiving your revised manuscript.

Kind regards,

Taiwo Opeyemi Aremu, MD, MPH, PhD

Academic Editor

PLOS ONE

Journal Requirements:

2. We note that this data set consists of interview transcripts. Can you please confirm that all participants gave consent for interview transcript to be published?

If they DID provide consent for these transcripts to be published, please also confirm that the transcripts do not contain any potentially identifying information (or let us know if the participants consented to having their personal details published and made publicly available). We consider the following details to be identifying information:

- Names, nicknames, and initials

- Age more specific than round numbers

- GPS coordinates, physical addresses, IP addresses, email addresses

- Information in small sample sizes (e.g. 40 students from X class in X year at X university)

- Specific dates (e.g. visit dates, interview dates)

- ID numbers

Or, if the participants DID NOT provide consent for these transcripts to be published:

- Provide a de-identified version of the data or excerpts of interview responses

- Provide information regarding how these transcripts can be accessed by researchers who meet the criteria for access to confidential data, including:

a) the grounds for restriction

b) the name of the ethics committee, Institutional Review Board, or third-party organization that is imposing sharing restrictions on the data

c) a non-author, institutional point of contact that is able to field data access queries, in the interest of maintaining long-term data accessibility.

d) Any relevant data set names, URLs, DOIs, etc. that an independent researcher would need in order to request your minimal data set.

For further information on sharing data that contains sensitive participant information, please see: https://journals.plos.org/plosone/s/data-availability#loc-human-research-participant-data-and-other-sensitive-data

If there are ethical, legal, or third-party restrictions upon your dataset, you must provide all of the following details (https://journals.plos.org/plosone/s/data-availability#loc-acceptable-data-access-restrictions):

a. A complete description of the dataset

b. The nature of the restrictions upon the data (ethical, legal, or owned by a third party) and the reasoning behind them

c. The full name of the body imposing the restrictions upon your dataset (ethics committee, institution, data access committee, etc)

d. If the data are owned by a third party, confirmation of whether the authors received any special privileges in accessing the data that other researchers would not have

e. Direct, non-author contact information (preferably email) for the body imposing the restrictions upon the data, to which data access requests can be sent

Reviewers' comments:

Reviewer's Responses to Questions

**Comments to the Author**

1. Is the manuscript technically sound, and do the data support the conclusions?

Reviewer #1: Yes

Reviewer #2: Partly

2. Has the statistical analysis been performed appropriately and rigorously?

Reviewer #1: N/A

Reviewer #2: N/A

3. Have the authors made all data underlying the findings in their manuscript fully available?

Reviewer #1: Yes

Reviewer #2: Yes

4. Is the manuscript presented in an intelligible fashion and written in standard English?

Reviewer #1: Yes

Reviewer #2: Yes

Reviewer #1: This manuscript provides a valuable qualitative inquiry into the lived experiences of individuals with long Covid in New Zealand, focusing on an important gap in NZ’s primary care system. Below are the comments to be addressed:

1. Convenience sampling from an online support group may introduce selection bias. Individuals who are active in such forums may be more health-literate or or having more severe symptoms than the general long Covid population.

2. Majority of the participants holding postgraduate degrees, limiting representativeness. It has been listed as one of the limitations of the current study. The authors shall put more emphasis on how this may influence self-advocacy and health system navigation is warranted.

3. The absence of Asian participants raises questions about inclusiveness.

4. The low number of male participants (3 out of 18) may also skew findings.

5. There is limited discussion on whether thematic saturation was reached.

6. The study is over-reliant on subjective interpretation. While narrative inquiry values lived experience, stronger triangulation—such as linking findings to provider perspectives or health system data—would enrich the analysis. The current presentation occasionally risks anecdotal generalisation.

7. Some participant quotes use emotive or colloquial expressions (e.g., “they haven’t known what the ***k to do with me”), which, while authentic, shall be better paraphrased for clarity and professionalism in a peer-reviewed academic journal.

8. The term “gaslighting” is used throughout without clear definition.

9. Implementation science frameworks (e.g., CFIR, RE-AIM) can be used to structure questions and recommendations.

Reviewer #2: Thank you for the invite and opportunity to review this important study regarding long COVID-19 sufferers and their experiences in NZ from a primary healthcare access perspective. The paper included important findings on a neglected area with potentially life changing implications for policy, practice and primary health care service delivery. I also thank the authors for a very readable and accessible manuscript. However, at its current form, there are key unaddressed points in different sections that warrant major revisions.

I attached my comments in a separate file for your information.

**Do you want your identity to be public for this peer review?** For information about this choice, including consent withdrawal, please see our Privacy Policy

Reviewer #1: **Yes: ** YIN-CHENG LIM

Reviewer #2: **Yes: ** Tamara Al-Obaidi

---

## [Author Response · Author response to Decision Letter 1]

24 Sep 2025

2.Participants in the study did not give consent for their interview transcripts to be published in entirety although they did consent to excerpts of the dataset being published as part of the qualitative coding process. All data excerpts used have been deidentified. The complete deidentified transcripts can be made available on request to the corresponding author.

---

## [Decision Letter · Decision Letter 1]

9 Oct 2025

Dear Dr. Rhodes,

Thank you for submitting your manuscript to PLOS ONE. After careful consideration, we feel that it has merit but does not fully meet PLOS ONE’s publication criteria as it currently stands. Therefore, we invite you to submit a revised version of the manuscript that addresses the points raised during the review process.

We look forward to receiving your revised manuscript.

Kind regards,

Taiwo Opeyemi Aremu, MD, MPH, PhD

Academic Editor

PLOS ONE

Journal Requirements:

Reviewers' comments:

Reviewer's Responses to Questions

**Comments to the Author**

Reviewer #1: All comments have been addressed

Reviewer #2: All comments have been addressed

2. Is the manuscript technically sound, and do the data support the conclusions?

Reviewer #1: Yes

Reviewer #2: Yes

3. Has the statistical analysis been performed appropriately and rigorously?

Reviewer #1: Yes

Reviewer #2: N/A

4. Have the authors made all data underlying the findings in their manuscript fully available?

Reviewer #1: Yes

Reviewer #2: Yes

5. Is the manuscript presented in an intelligible fashion and written in standard English?

Reviewer #1: Yes

Reviewer #2: Yes

Reviewer #1: Thanks for the invitation to review the first revised version of this manuscript on “Experiences of accessing primary care by those living with long Covid in New Zealand: a qualitative analysis”. Most of the comments from the reviewers have been addressed satisfactorily. However, there are few questions below to be addressed.

1. There is limited reflexivity statement within the “Methods” section. The authors did not include professional background, prior experience with long Covid in this component. This omission limits readers’ ability to assess how the researchers’ positions may have influenced data collection, interpretation, and thematic analysis.

2. The limitations acknowledge selection bias and overrepresentation of highly educated participants. However, there is no reflection on how this bias may have shaped thematic dominance (e.g., emphasis on self-advocacy). This may limits the transferability to underrepresented groups, particularly Māori and Pacific communities.

3. There is misalignment between abstract conclusion and main conclusion. The abstract conclusion mentions “creative solutions for primary care,” whereas the main conclusion emphasises government investment and allied health utilisation.

4. Minorities-Māori and Pacific participants are represented. However, their voices are not distinctly analysed within the results. This may lead to the missed opportunity to discuss health inequities central to the New Zealand context.

Reviewer #2: I thank the authors for addressing each comment and for adding more information to a satisfactory level.

**Do you want your identity to be public for this peer review?** For information about this choice, including consent withdrawal, please see our Privacy Policy

Reviewer #1: **Yes: ** YIN-CHENG LIM

Reviewer #2: **Yes: ** Dr Tamara Al-Obaidi

---

## [Author Response · Author response to Decision Letter 2]

12 Oct 2025

Please see the uploaded Response to Reviewers file

---

## [Decision Letter · Decision Letter 2]

21 Oct 2025

Experiences of accessing primary care by those living with long Covid in New Zealand: a qualitative analysis

PONE-D-25-18859R2

Dear Dr. Rhodes,

We’re pleased to inform you that your manuscript has been judged scientifically suitable for publication and will be formally accepted for publication once it meets all outstanding technical requirements.

Kind regards,

Taiwo Opeyemi Aremu, MD, MPH, PhD

Academic Editor

PLOS ONE

Additional Editor Comments (optional):

Reviewers' comments:

Reviewer's Responses to Questions

**Comments to the Author**

Reviewer #1: All comments have been addressed

2. Is the manuscript technically sound, and do the data support the conclusions?

Reviewer #1: Yes

3. Has the statistical analysis been performed appropriately and rigorously?

Reviewer #1: Yes

4. Have the authors made all data underlying the findings in their manuscript fully available?

Reviewer #1: Yes

5. Is the manuscript presented in an intelligible fashion and written in standard English?

Reviewer #1: Yes

Reviewer #1: The authors have addressed all comments satisfactorily for the revision 1 and revision 2. I have no further comments to make.

**Do you want your identity to be public for this peer review?** For information about this choice, including consent withdrawal, please see our Privacy Policy

Reviewer #1: **Yes: ** YIN CHENG LIM

---

## [Editor Report · Acceptance letter]

PONE-D-25-18859R2

PLOS ONE

Dear Dr. Rhodes,

I'm pleased to inform you that your manuscript has been deemed suitable for publication in PLOS ONE. Congratulations! Your manuscript is now being handed over to our production team.

Kind regards,

on behalf of

Dr. Taiwo Opeyemi Aremu

Academic Editor

PLOS ONE